# Effect of Attapulgite Application on Aggregate Formation and Carbon and Nitrogen Content in Sandy Soil

Ziru Niu [1,2], Yongzhong Su [3,*], Juan Li [1,2], Fangjiao An [4] and Tingna Liu [3]

1    Shaanxi Provincial Land Engineering Construction Group, Key Laboratory of Degraded and Unused Land Consolidation Engineering, Ministry of Natural Resources, Xi'an 710000, China; niuziru12345@163.com (Z.N.); lijuan8136@163.com (J.L.)
2    Shaanxi Engineering Research Center of Land Consolidation, Shaanxi Provincial Land Consolidation Engineering Technology Research Center, Xi'an 710000, China
3    Key Laboratory of Ecological Safety and Sustainable Development in Arid Lands, Northwest Institute of Eco-Environment and Resources, Chinese Academy of Sciences, Lanzhou 730000, China; liutn@lzb.ac.cn
4    School of Civil Engineering, Lanzhou University of Technology, Lanzhou 730000, China; anfj@lzb.ac.cn
*    Correspondence: suyzh@lzb.ac.cn

**Abstract:** Clay minerals are the main cementing substances for sandy soils to form aggregates. The clay mineral attapulgite clay is abundant in Northwest China, and its special colloidal properties and crystal structure make it excellent in improving soil physicochemical properties. Using attapulgite as soil conditioner, the effects of different application rates of attapulgite on the formation and stability of sandy soil aggregates were studied through field experiments for two consecutive years. The results showed that the application of 6000 kg·hm$^{-2}$ attapulgite soil in sandy soil farmland for two consecutive years reduced the soil bulk density by 0–20 cm, from 1.55 g·cm$^{-3}$ to 1.47 g·cm$^{-3}$, a decrease of 3.6%; the soil pH was increased by 3.7% from 8.59 to 8.84. The soil organic carbon, inorganic carbon and total nitrogen in the whole soil increased by 4.52%, 5.23% and 6.22%, respectively. The mass fraction of macro-aggregates of 2–0.25 mm and micro-aggregates of 0.25–0.053 mm as well as the contents of organic carbon, inorganic carbon and total nitrogen increased by 3.5%, 5.2%, 8.7%, 5.6% and 6.7%, respectively, thus improving the stability of aggregates. However, low application rates (1500 kg·hm$^{-2}$ and 3000 kg·hm$^{-2}$) of attapulgite had no significant effect on soil physical and chemical properties. Attapulgite, as a kind of highly adsorptive clay mineral, can be directly applied to sandy soil to increase soil cementitious substances, promote the formation of soil aggregates and increase the carbon and nitrogen fixation capacity of sandy soil. The improvement effect on the formation and stability of aggregates will gradually accumulate with the years of application. Therefore, in the future, the effects of adding attapulgite on the growth of various crops under various types of soil and climatic conditions should be carried out to obtain more systematic conclusions.

**Keywords:** attapulgite; aggregates; aggregate stability; carbon and nitrogen content



## 1. Introduction

Agriculture is the guarantee for human survival, and it continuously provides food and basic material guarantee for human life [1–3]. Agricultural development plays an important role in all countries in the world, and its development status directly affects the overall economic development of all countries [4]. China is a country based on agriculture, which plays a central role in its development. In arid and semi-arid regions, the phenomenon of soil degradation and land desertification has attracted people's attention, especially the Hexi Corridor region in northwest China, which is the main oasis for agricultural region in northwest China, and its soil quality affects the sustainable development of local agriculture [5–7]. The farmland reclaimed in Zhangye Oasis in the Hexi Corridor in the past 30 years accounts for more than one third of the farmland area of the oasis [8]. The natural soil of the newly reclaimed farmland is gray-brown desert soil, gray desert soil, aeolian

sandy soil and calcareous soil, most of which are sandy soil or shallow soil containing relatively coarse sand [9]. The weak ability of water and fertilizer conservation is the main problem restricting agricultural production. Due to the construction of the upstream reservoir and the improvement of the farmland canal system, the surface water irrigation input of fine sand into the soil is reduced [10]. The application of traditional farmyard manure (cow dung) adds a large amount of fine matter to the farmland soil. Because the modern aquaculture industry has been gradually separated from the farmland nutrient circulation system, it is difficult to form a fertile soil with good structure in the old oasis due to the lack of input of clay, silt and clay, the lack of cementitious materials that form soil aggregates and the slow development of artificial soil [11].

Attapulgite (ATP), also known as palygorskite or palygorskite, is a natural hydrous magnesium-aluminosilicate two-to-one clay mineral with nano-rod crystal structure, with strong adsorption, colloid and colloid properties. Attapulgite clay minerals can play a good role as nutrient carriers in soil. This is mainly due to the adsorbability and ion exchange capacity of clay minerals, allowing them to effectively adsorb and retain nutrients in the soil, including cations and anions. This adsorption and ion exchange ability enables clay minerals to act as a stable reservoir of nutrients in the soil, reducing nutrient loss and leaching, which is conducive to the absorption and utilization of nutrients by plants [12]. Due to its large specific surface area and strong adsorption capacity, attapulgite can firmly adsorb the soluble heavy metal elements in the soil on the surface of the clay or implant them into the deep soil chain structure. Moreover, attapulgite contains a certain amount of $Mg^{2+}$, $Fe^{3+}$ and other metal ions and has strong cation exchange capacity, which has a good effect on the remediation of heavy metal contaminated soil [13,14]. The fibrous network structure and strong adsorption of attapulgite can be used as a good sustained-release carrier, which has been applied in the production of slow-release fertilizer. At the same time, it is also a cementitious material for granulation in the production of compound fertilizer or a slow-release or controlled-loss fertilizer coating agent [15,16].

The quantity and quality of soil aggregates are important indicators of soil structure and play a crucial role in ecosystem functioning, affecting soil moisture, air, nutrient flow and storage and microbial activity [17]. It was found that the biological crust succession had significant influence on the stability of surface soil aggregates during the ecological project of "returning farmland to forest" on the Loess Plateau [18]. Wei's results show that the increase of soil aggregates during vegetation restoration in karst areas is significantly correlated with the increase of nutrient content [19]. In the arid northwest region, the soil organic matter content is relatively low, and there is a lack of cemented substances such as clay particles [20–23]. Attapulgite soil is directly added to silty soil; because of its significant gelling characteristics, the dispersed particles can be heavy into larger particle aggregates, the average particle size of soil aggregates is increased, and the water holding performance is improved, which should have a good effect on the development of sandy soil structure, the improvement of water holding and fertilizer retention performance and the reduction of fertilizer nutrient leaching loss [24]. Wang prepared environmentally friendly composite superabsorbent sand fixing materials using the acid modification process from attapulgite soil, which improved the stability of dry aggregates in sandy soil [25]. Luo found that attapulgite slow-release fertilizer can effectively adsorb or distribute soil water and nutrients in the network structure, effectively reducing the leaching loss of water and nutrients [26]. The field results of Naraghi showed that attapulgite soil prepared as phosphatediamine and urea slow-release fertilizer can significantly increase the yield of wheat, potato and some Chinese medicinal materials [27]. The results of the above scholars show that attapulgite clay minerals have fiber network structure and significant gelling characteristics, and direct application or as functional slow-release (controlled) fertilizer application in sandy soil has a great effect on the development of sandy soil structure, the improvement of water holding and fertilizer retention performance and the reduction of fertilizer nutrient leaching loss, but the development and application of attapulgite soil is still in its infancy.

Aeolian sand is widely distributed in northwest China, with low organic matter content and difficulty to accumulate, high sand content, poor soil structure and general lack of nutrients. The improvement of soil fertility and carbon fixation capacity has become the top priority of local agricultural development [28]. The Linze area in the middle of the Hexi Corridor has very rich mineral resources of attapulgite clay [29]. In recent decades, due to the limited application amount of organic fertilizer and the reduction of surface water irrigation area, the input of soil fine matter is blocked. Under this background, the newly reclaimed sandy soil structure is slowed down, and the soil texture becomes more loose, which leads to the decline of its water and fertilizer retention ability, affecting the normal growth and yield of crops. In this context, effective soil improvement measures, such as the rational use of soil amendments such as attapulgite, can help to promote the formation of soil structure, improve soil fertility and water retention, thereby increasing crop yield and achieving sustainable agricultural development [30]. The purpose of this study is to rapidly change the texture of sandy soil, promote the formation of soil aggregates and increase the carbon sequestration and nitrogen fixation capacity of sandy soil by the application of attapulgite clay. Our hypothesis is that improving sandy soil by applying attapulgite clay can effectively improve soil texture and structure, thereby increasing soil carbon and nitrogen fixation capacity. Our expected objectives are (1) that the application of chemical fertilizers alone has no significant effect on the physical and chemical properties of sandy soils and (2) that the physical and chemical properties of sandy soils gradually change as the application rate of attapulgite soil increases.

## 2. Materials and Methods

### 2.1. Overview of the Study Area

The research area is located in the oasis zone to the north of Linze in the middle of Hexi Corridor in Gansu Province (Figure 1). It is a newly developed oasis (39°24′ N, 100°21′ E, altitude 1350–1385 m). In recent decades, it has gradually expanded from the old oasis to the desert edge. Oasis is surrounded by desert and Gobi, with typical desert climate. In recent years, the average precipitation is 116.8 mm, the annual evaporation is 2390 mm, the annual average temperature is 7.6 °C, and the frost-free period is 165 days. The zonal soil is gray brown desert soil. Due to the long-term invasion and sedimentation of aeolian sand, aeolian sand was formed in the northeast of the oasis edge. Since the mid-1970s, the desertification control in the marginal area of the oasis has been carried out, and the desert has been continuously reclaimed and cultivated, gradually forming sandy irrigation farmland with different reclamation sequences. Before 2000, farmland was irrigated by surface water from the Heihe River. After 2000, groundwater from the lower reaches of the Heihe River was mainly used to irrigate farmland, accounting for more than half of the irrigation water.

In the 1980s, ecological migration was carried out in the gray-brown desert soil distribution area in the north of the oasis (an engineering village) and gradually began to reclaim farmland for oasis irrigation. Since the groundwater is alkaline water, the reclaimed farmland is all irrigated with surface water. In recent decades, seed corn and field corn are the main crops in this area, which are cultivated by traditional farming and plastic film mulching. Due to the poor structure and low nutrient content of sandy soil, its water and fertilizer conservation effect is relatively poor, and it is the area with the largest water consumption and fertilizer application in oasis irrigation in arid areas. In order to plant corn, the amount of fertilizer required to be applied each year is 90–150 kg·hm$^{-2}$ for nitrogen fertilizer, 90–150 kg·hm$^{-2}$ for phosphorus fertilizer and 60–90 kg·hm$^{-2}$ for potassium fertilizer. In addition, about 6–11 times of irrigation are required during the growth cycle of corn.

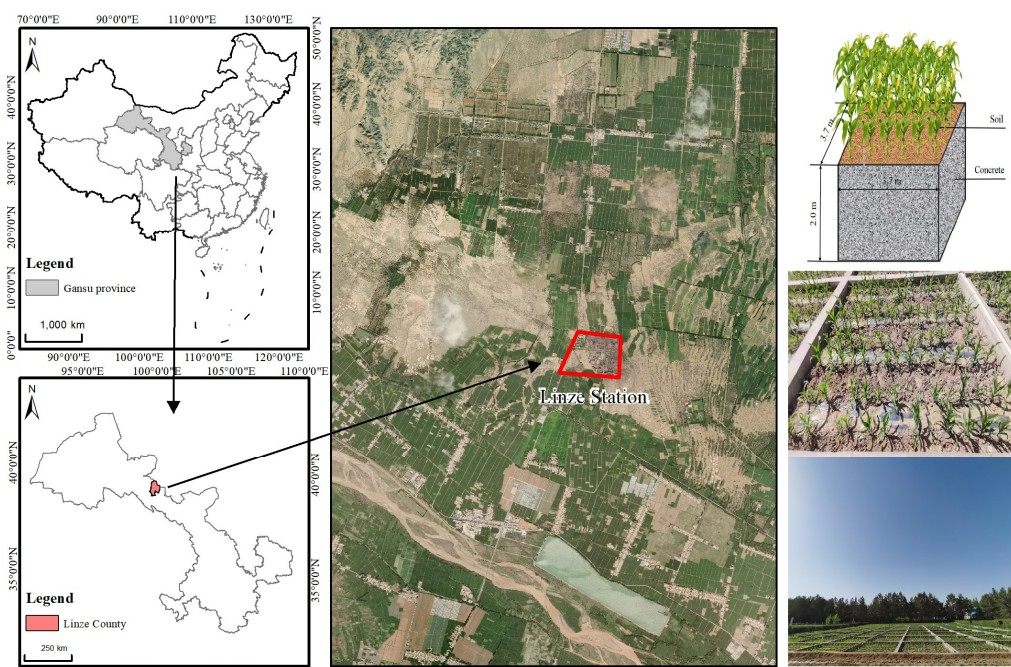

**Figure 1.** Location map of attapulgite positioning test.

## 2.2. Experiment Design

From 2020 to 2021, a field experiment of applying attapulgite to maize in sandy farmland was arranged in the farmland test field of Linze Station (Figure 1). The test soil was irrigated and cultivated into sandy newly formed soil (irrigated and cultivated aeolian sandy soil), and the profile properties are shown in Table 1. The experiment set up 6 treatments: Teatment 1 (T1), no fertilization + no application of attapulgite clay control (CK); Treatment 2 (T2), conventional fertilization + no application of attapulgite clay (NPK1), Treatment 3 (T3): conventional fertilization + application of attapulgite clay 1500 kg·hm$^{-2}$ (NPK1A1); Treatment (T4), conventional fertilization + application of attapulgite clay 3000 kg·hm$^{-2}$ (NPK1A2); Treatment 5 (T5), conventional fertilization + application of attapulgite clay 6000 kg·hm$^{-2}$ (NPK1A4); Treatment 6 (T6), 20% reduction of chemical fertilizer + 6000 kg·hm$^{-2}$ application of attapulgite clay (NPK0.8A4). The conventional fertilization level was N-P$_2$O$_5$-K$_2$O, 320–240–200 kg·hm$^{-2}$; the 20% reduction of fertilizer application level was N-P$_2$O$_5$-K$_2$O, 256–192–160 kg·hm$^{-2}$. The fertilizer applied was attapulgite-based compound fertilizer, which was granulated with attapulgite clay mixed with urea, diammonium phosphate and potassium sulfate in a certain proportion. The nutrient content was N-P$_2$O$_5$-K$_2$O-P$_2$O$_5$ 17.8%-16.1%-10%, containing 20% attapulgite clay, and the total nutrient content was 43.9%. The organic carbon content of attapulgite clay minerals was 5.43 g·kg$^{-1}$, the inorganic carbon content was 6.71 g·kg$^{-1}$, the total nitrogen content was 0.67 g·kg$^{-1}$, and the pH was 7.9. One-time light and simplified fertilization method were adopted, all of which were used for basic application. Attapulgite-based compound fertilizer and urea were combined to provide the required N, P and K total nutrients. Before sowing, after the mixture of attapulgite clay and fertilizer was evenly spread, the soil was lightly turned and covered with plastic film. Six treatments were repeated for 3 times in 18 cells with a cell area of 20 m$^2$ (4 m × 5 m) and random block design.

**Table 1.** Soil physical and chemical properties of the experiment fields.

| Soil Layer | Sand (%) | Silt (%) | Clay (%) | BD (g·cm$^{-3}$) | SOM (g·kg$^{-1}$) | TN (g·kg$^{-1}$) | pH |
|---|---|---|---|---|---|---|---|
| 0–20 cm | 81.6 | 8.2 | 10.3 | 1.55 | 5.47 | 0.36 | 8.90 |
| 20–40 cm | 89.0 | 5.2 | 5.9 | 1.54 | 3.69 | 0.22 | 9.10 |
| 40–60 cm | 90.7 | 3.1 | 6.2 | 1.57 | 2.11 | 0.17 | 9.05 |
| 60–80 cm | 91.9 | 2.4 | 5.7 | 1.62 | 1.88 | 0.11 | 9.15 |
| 80–100 cm | 91.9 | 2.4 | 5.8 | 1.62 | 1.72 | 0.11 | 9.15 |

BD: soil bulk density; SOM: soil organic matter; TN: soil total nitrogen.

The depth of 0–100 cm between the communities was separated by waterproof materials. The tested maize variety was Yudan 9932. On 15 September 2020 and 20 September 2021, respectively, after corn harvest, 0–20 cm topsoil samples were taken from plots to determine the formation of soil aggregates and the change of carbon and nitrogen content after the application of attapulgite clay.

*2.3. Aggregate Separation*

For each field, the plum blossom method was used for sampling. First, the soil surface was cleaned, and then the undisturbed soil was removed with a small shovel. During the sampling process, the undisturbed soil should avoid extrusion deformation. Then, the soil clumps at five sample sites were mixed. Soil samples were physically separated into water-stable aggregates of different particle sizes using the wet sieve method according to Elliott [31]. The test was performed as follows: the sets of sieves were arranged in order of pore size (2 mm, 0.25 mm and 0.053 mm) on an oscillating stand and placed in a bucket of water. The height of the water in the bucket was adjusted so that the upper edge of the top sieve was slightly above the water surface when the oscillation was at the lowest position, and the soil was still immersed in water when the oscillation was at the highest position. A 50 g soil sample was placed in the top sieve of the 2 mm, 0.25 mm and 0.053 mm diameter sieves, pre-saturated in distilled water for 3 min, and then shaken vertically in water 40 times (3 cm amplitude) at a rate of 15 oscillations per minute. The aggregates were collected from each layer of the nested sieve, transferred to a beaker, dried at 60 °C, and weighed. Since sandy soil aggregates contain a large amount of sand particles, the mass fraction of aggregates was corrected using the method of Six [32,33]. The previously weighed aggregates were transferred to a beaker; sodium hexametaphosphate was added to separate the aggregates from the sand particles; then, the sand particles were removed, and the remaining aggregates were transferred to a beaker, dried and finally weighed to obtain the accurate mass fraction of the aggregates.

*2.4. Determination of Soil Physical and Chemical Properties*

The soil particle size composition was determined using the wet sieve plus pipette method [34], the total carbon and total nitrogen content of soil and aggregates of different particle sizes were determined with the element analyzer (Elemental variable macro cube, Germany), and the organic carbon was determined using the Walkley–Black potassium dichromate oxidation method [35]. The soil pH and conductivity (EC) were measured directly by pH meter. The content of soil inorganic carbon (SIC) was obtained by subtracting organic carbon from total carbon.

*2.5. Data Analysis*

Calculation of the average mass diameter of soil aggregates is as follows:

$$MWD = \sum Xi \cdot Wi$$

where MWD is the average mass diameter of aggregates (mm), Xi is the average diameter of aggregates (mm) in any class range, and Wi is the percentage of aggregates corresponding to Wi.

$$\text{Contribution rate of aggregate SOC} = \frac{\text{Ni} \times \text{Wi}}{\sum_1^n (\text{Ni} \times \text{Wi})}$$

where Ni is the organic carbon content of aggregates of each particle size ($g \cdot kg^{-1}$), and Wi is the proportion of aggregate mass of each particle size.

Microsoft Excel 2010 and origin18 were used for data processing, statistics and plots. Data measurements were expressed as mean $\pm$ standard deviation. Significance of differences between treatments was analyzed using one-way ANOVA ($\alpha$ = multiple mean difference test (LSD 0.05)) with SPSS 22 software. Pearson correlation analysis was used to analyze the correlation of the results of the normality test data. Amos 16 software was used as a structural equation model (SEM) to simulate the effects of mineral clay amendments on soil structure (MWD) and carbon and nitrogen content.

## 3. Results

### 3.1. Effect of Application of Attapulgite on Physical and Chemical Properties of Aeolian Sand Soil

Compared with the control group, the physicochemical properties of the surface (0–20 cm) soil in each treatment changed to a certain extent after the application of attapulgite soil (Table 2). After two consecutive years of trials, the application of high doses (6000 $kg \cdot hm^{-2}$) of attapulgite (NPK1A4 and NPK0.8A4) had a significant effect on soil volume ($p < 0.05$). In the first year (2020) of the experiment, with the increase in attapulgite soil application, although the soil bulk density of each community showed a decreasing trend as a whole, there was no significant difference. In the second year of the experiment, the soil bulk density showed a significant decrease trend, from 1.55 $g \cdot cm^{-3}$ in the control group to 1.47 $g \cdot cm^{-3}$, a decrease of 5.16%. The application of attapulgite soil for two consecutive years gradually changed the soil granular composition in all treatments—that is, the proportion of sand fractions gradually decreased from 75.56% in 2020 to 68.56% and from 76.23% in 2021 to 68.47%, and the proportion of silt + clay gradually increased, from 24.44% in 2020 to 31.24% and from 23.77% in 2021 to 30.82%, respectively.

**Table 2.** Effects of attapulgite application on soil physical and chemical properties.

| | Treatments | Treatment | | | | | | ANOVA | |
| | | CK | NPK1 | NPK1A1 | NPK1A2 | NPK1A4 | NPK0.8A4 | F | p |
|---|---|---|---|---|---|---|---|---|---|
| 2020 year | BD (g cm$^{-3}$) | 1.55 ± 0.01 a | 1.54 ± 0.08 a | 1.55 ± 0.04 a | 1.54 ± 0.06 a | 1.51 ± 0.04 a | 1.50 ± 0.03 a | 2.26 | >0.05 |
| | Sand (%) | 75.56 ± 1.31 a | 76.85 ± 2.14 a | 75.61 ± 1.19 ab | 72.33 ± 1.34 b | 69.97 ± 1.18 b | 68.56 ± 1.02 b | 16.99 | <0.001 |
| | Silt (%) | 13.31 ± 0.74 c | 10.59 ± 0.88 d | 11.43 ± 1.07 b | 14.46 ± 0.54 b | 14.71 ± 0.58 c | 14.23 ± 0.46 a | 94.913 | <0.001 |
| | Clay (%) | 11.13 ± 0.87 b | 12.56 ± 0.65 ab | 12.96 ± 0.51 b | 13.21 ± 0.63 a | 15.32 ± 0.65 a | 17.21 ± 0.86 a | 7.46 | <0.01 |
| | pH | 8.58 ± 0.12 b | 8.49 ± 0.06 b | 8.72 ± 0.04 a | 8.74 ± 0.07 a | 8.75 ± 0.13 a | 8.79 ± 0.10 a | 41.45 | <0.001 |
| | EC (μs·cm$^{-1}$) | 178 ± 9.8 c | 189 ± 7.5 a | 174 ± 12 a | 197 ± 9.2 a | 182 ± 12 b | 178 ± 7.0 b | 16.30 | <0.001 |
| | SOC (g·kg$^{-1}$) | 5.28 ± 0.44 c | 5.16 ± 0.36 c | 5.25 ± 0.31 c | 5.72 ± 0.10 b | 6.17 ± 0.09 a | 6.25 ± 0.16 a | 34.28 | <0.001 |
| | SIC(g·kg$^{-1}$) | 3.96 ± 0.24 d | 4.12 ± 0.29 c | 3.76 ± 0.07 d | 4.43 ± 0.33 b | 4.86 ± 0.18 b | 4.93 ± 0.25 a | 19.10 | <0.001 |
| | TN (g·kg$^{-1}$) | 0.39 ± 0.01 c | 0.37 ± 0.03 c | 0.35 ± 0.03 b | 0.42 ± 0.02 c | 0.54 ± 0.06 c | 0.55 ± 0.02 a | 6.56 | <0.01 |
| 2021 year | BD (g cm$^{-3}$) | 1.55 ± 0.05 a | 1.55 ± 0.04 ab | 1.54 ± 0.01 b | 1.53 ± 0.04 ab | 1.48 ± 0.07 ab | 1.47 ± 0.05 ab | 8.5 | <0.01 |
| | Sand(%) | 76.23 ± 0.49 a | 75.85 ± 1.89 a | 74.80 ± 1.14 ab | 72.43 ± 1.34 b | 68.47 ± 1.69 b | 69.18 ± 1.78 b | 36.61 | <0.001 |
| | silt(%) | 10.56 ± 0.84 c | 12.37 ± 0.93 b | 11.28 ± 1.47 b | 12.17 ± 1.36 a | 13.64 ± 1.36 b | 13.58 ± 1.22 a | 21.75 | <0.001 |
| | clay(%) | 13.21 ± 0.54 | 11.78 ± 0.18 | 13.92 ± 0.32 | 15.4 ± 0.75 | 16.89 ± 0.42 | 17.24 ± 0.96 | 7.65 | <0.01 |
| | pH | 8.56 ± 0.08 b | 8.62 ± 0.08 b | 8.75 ± 0.11 a | 8.71 ± 0.09 a | 8.78 ± 0.14 a | 8.75 ± 0.14 a | 38.18 | <0.001 |
| | EC (μs·cm$^{-1}$) | 167 ± 5.5 c | 176 ± 7.0 b | 188 ± 9.2 a | 183 ± 6.5 a | 177 ± 3.4 b | 165 ± 8.6 c | 28.53 | <0.001 |
| | SOC (g·kg$^{-1}$) | 5.58 ± 0.19 c | 5.86 ± 0.36 b | 5.83 ± 0.28 b | 5.96 ± 0.56 b | 6.57 ± 0.28 a | 6.20 ± 0.18 a | 45.78 | <0.001 |
| | SIC(g·kg$^{-1}$) | 3.71 ± 0.10 d | 4.44 ± 0.48 c | 4.10 ± 0.49 c | 4.66 ± 0.09 c | 5.49 ± 0.24 b | 5.63 ± 0.19 a | 32.04 | <0.001 |
| | TN (g·kg$^{-1}$) | 0.32 ± 0.04 b | 0.39 ± 0.12 ab | 0.38 ± 0.12 ab | 0.45 ± 0.10 a | 0.64 ± 0.06 a | 0.62 ± 0.07 a | 18.88 | <0.001 |

Note: BD: soil bulk density; EC: electrical conductivity; SOC: soil organic carbon; SIC: soil inorganic carbon; TN: soil total nitrogen. The value is the mean $\pm$ 1 standard deviation of three replicate samples. Indicated with different lowercase letters, the difference is statistically significant ($p < 0.05$).

The application of attapulgite significantly increased the pH of aeolian sand compared to the control group (Table 2). In 2020 and 2021, the pH of aeolian sand showed an increasing trend with the increase of attapulgite application. Compared with the control group, the

soil pH increased from 8.58 to 8.88 in the first year (2020) trial from 8.58 to 8.88, an increase of 3.50%, and the soil pH in the NPK1A4 treatment in the second year (2021) trial increased from 8.56 to 8.85, an increase of 3.39%. The contents of organic carbon, inorganic carbon and total nitrogen in sandy soil showed the same trend as pH and increased with the increase of attapulgite application. After two years of application, the increase rates of soil organic carbon, inorganic carbon and total nitrogen in the whole soil were 18.37%, 24.49% and 41.03%, respectively. However, the application of chemical fertilizer alone and the attapulgite soil with low application rate (1500 kg·hm$^{-2}$) had no significant effect on the physical and chemical properties of sandy soil.

### 3.2. Effect of Application of Attapulgite on Aggregate Distribution and Stability

Different doses of attapulgite soil application had a certain effect on the distribution of aeolian sandy aggregates (Table 3). Compared with CK, the application of chemical fertilizer alone (NPK1) and low-application attapulgite soil in 2020 and 2021 had no significant effect on the particle level composition of each aggregates ($p > 0.05$), while the proportion of macro-aggregates (2–0.25 mm) was significantly increased by high-application attapulgite. Compared with CK, NPK1A4 and NPK0.8A4 increased the mass fraction of 2–0.25 mm granular aggregates by 29.80% and 32.81%, respectively, and the increase in 0.25–0.053 mm micro-aggregates was consistent, increasing by 71.47% and 73.35%, respectively. The mass fraction of >2 mm macro-aggregates did not change significantly. In addition, the average weight diameter (MWD) of aggregates in each treatment was also affected by the application amount of attapulgite soil, and the order of change was NPK0.8A4 = NPK1A4 > NPK1A2 > NPK1A1 > NPK1 = CK, where the NPK0.8A4 and NPK1A4 average weight diameter (MWD) of the aggregates was the maximum in the two consecutive years of experiments, both of which were 0.22, which was significantly higher than that of other treatments.

**Table 3.** Distribution of aggregate fractions for different cultivation ages.

| | | Treatment | | | | | | ANOVA | |
|---|---|---|---|---|---|---|---|---|---|
| | | CK | NPK1 | NPK1A1 | NPK1A2 | NPK1A3 | NPK1A4 | F | p |
| 2020 | >2 mm | 4.45 ± 0.05 a | 4.27 ± 0.04 a | 4.79 ± 0.06 a | 4.62 ± 0.16 a | 4.49 ± 0.39 a | 4.66 ± 0.25 a | 1.91 | >0.05 |
| | 0.25–2 mm | 10.33 ± 0.30 c | 11.62 ± 0.50 b | 11.81 ± 0.62 b | 12.02 ± 0.31 b | 13.41 ± 0.42 a | 14.72 ± 0.17 a | 23.44 | <0.001 |
| | 0.25–0.053 mm | 3.19 ± 0.27 c | 3.35 ± 0.53 c | 4.57 ± 0.19 b | 5.34 ± 0.54 a | 5.47 ± 0.67 a | 5.32 ± 0.05 a | 22.58 | <0.001 |
| | <0.053 mm | 14.19 ± 0.61 a | 11.47 ± 0.59 c | 12.95 ± 0.69 b | 13.26 ± 0.52 a | 13.30 ± 1.17 a | 14.14 ± 1.01 a | 8.21 | <0.01 |
| | MWD | 16.55 ± 0.28 e | 17.85 ± 0.46 d | 18.77 ± 0.67 c | 18.95 ± 0.36 c | 20.41 ± 0.44 b | 22.03 ± 0.42 a | 32.23 | <0.001 |
| 2021 | >2 mm | 4.35 ± 0.50 b | 4.07 ± 0.34 c | 4.45 ± 0.32 b | 4.71 ± 0.19 a | 5.05 ± 0.20 a | 4.48 ± 0.09 b | 4.09 | <0.05 |
| | 0.25–2 mm | 10.86 ± 0.30 c | 11.74 ± 0.06 c | 12.25 ± 0.35 b | 13.25 ± 0.22 b | 13.99 ± 0.20 a | 14.74 ± 0.14 a | 54.88 | <0.001 |
| | 0.25–0.053 mm | 3.36 ± 0.58 c | 3.47 ± 0.39 c | 4.00 ± 0.87 b | 4.63 ± 0.36 b | 4.73 ± 0.45 b | 5.67 ± 0.38 a | 10.75 | <0.001 |
| | <0.053 mm | 12.40 ± 0.45 b | 12.70 ± 1.12 b | 13.79 ± 0.71 b | 14.72 ± 0.83 a | 14.71 ± 1.04 a | 15.31 ± 0.97 a | 7.14 | <0.01 |
| | MWD | 17.08 ± 0.37 e | 17.80 ± 0.36 d | 18.85 ± 0.05 c | 20.33 ± 0.40 b | 21.70 ± 0.27 a | 21.92 ± 0.22 a | 79.34 | <0.001 |

MWD: the mean weight diameter. Values are the average of values from five replicate samples ± one standard deviation. Different lowercase letters indicate significant differences ($p < 0.05$) among the different aggregate fractions.

### 3.3. Effect of Application of Attapulgite on Carbon and Nitrogen Content of Aggregates in Aeolian Sand Soil

The effects of separate application of fertilizer and attapulgite soil on the carbon and nitrogen content in aeolian sandy aggregates are shown in Figure 2. Compared with CK, the application of chemical fertilizer alone (NPK1) and low-application attapulgite in 2020 and 2021 had no significant effect on the granular level organic carbon, inorganic carbon, total nitrogen and carbon–nitrogen ratios of each aggregate ($p > 0.05$). In the first year (2020) of the experiment, the organic carbon content in >2 mm, 2–0.25 mm, 0.25–0.053 mm aggregates and silt + clay gradually increased with the increase in attapulgite application, from 6.68 g·kg$^{-1}$, 4.48 g·kg$^{-1}$, 2.4 g·kg$^{-1}$, 7.46 g·kg$^{-1}$ in CK to 7.46 g·kg$^{-1}$, 5.57 g·kg$^{-1}$, 3.67 g·kg$^{-1}$, 9.08 g·kg$^{-1}$ (NPK0.8A4), an increase of 11.67%, 24.33%, 52.92% and 21.72%. In the second year (2021) experiment, the organic carbon content in the aggregates increased more significantly; the organic carbon content in each particles aggregate increased with the increase in attapulgite application; and the organic carbon content in each par-

ticles aggregate increased from 6.73 g·kg$^{-1}$, 4.42 g·kg$^{-1}$, 2.41 g·kg$^{-1}$ and 7.33 g·kg$^{-1}$ to 7.78 g·kg$^{-1}$, 5.71 g·kg$^{-1}$, 3.76 g·kg$^{-1}$ and 9.13 g·kg$^{-1}$, respectively. The growth rates were 15.60%, 29.19%, 56.02% and 24.56%. The contents of inorganic carbon and total nitrogen in each particles aggregate increased significantly in the first year of the experiment (2020), from 4.70 g·kg$^{-1}$, 3.69 g·kg$^{-1}$, 4.89 g·kg$^{-1}$ and 5.78 g·kg$^{-1}$ in NPK0.8A4 to 6.24 g·kg$^{-1}$, 5.09 g·kg$^{-1}$, 6.04 g·kg$^{-1}$ and 7.69 g·kg$^{-1}$ in NPK0.8A4, with an increase of 23.52–37.94%. The total nitrogen content increased from 0.53 g·kg$^{-1}$, 0.45 g·kg$^{-1}$, 0.30 g·kg$^{-1}$ and 0.63 g·kg$^{-1}$ in CK to 0.67 g·kg$^{-1}$, 0.60 g·kg$^{-1}$, 0.43 g·kg$^{-1}$ and 0.78 g·kg$^{-1}$ in NPK0.8A4, with an increase of 23.8–43.33%. The carbon–nitrogen ratio in each particles agglomerate in all treatments showed a trend of first increasing and then decreasing with the application of attapulgite.

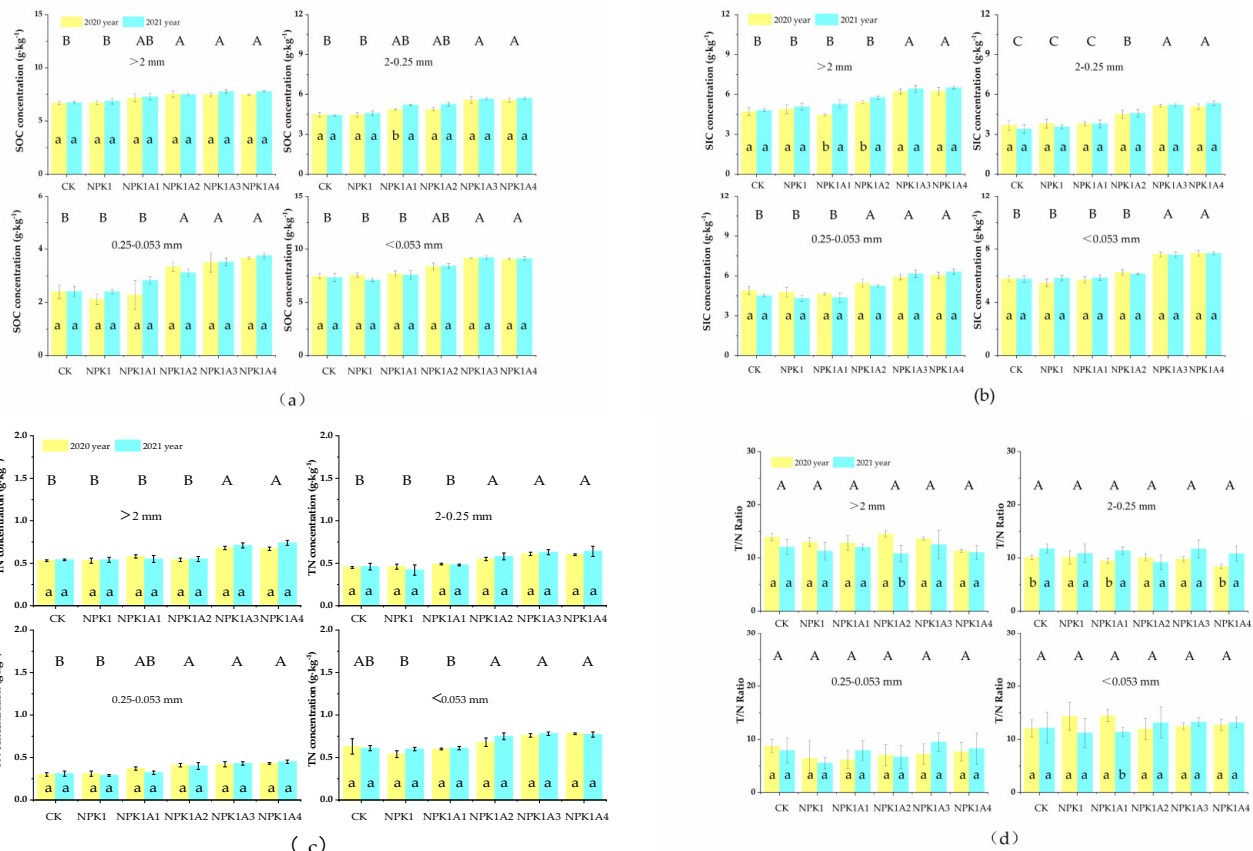

**Figure 2.** Effect of attapulgite application on carbon and nitrogen content in aggregates; (**a**) the effect of attapulgite application on the content of organic carbon in aggregates of each particle size; (**b**) the application of attapulgite on the content of inorganic carbon in aggregates of various particle sizes; (**c**) effect of attapulgite application on total nitrogen content in aggregates of each particle size; (**d**) effect of attapulgite application on carbon–nitrogen ratio in aggregates of various. The data in the figure are mean ± standard deviation, and different capital and lowercase letters indicate that the difference between different years under the same treatment is significant at the *p* < 0.05 level.

The application age of attapulgite has a certain effect on the carbon and nitrogen content in aeolian sandy aggregates (Figure 2). After comparing 2020 with 2021, it is found that the organic carbon, inorganic carbon, total nitrogen content and carbon–nitrogen ratio in each particles aggregates treated with high application of attapulgite soil have increased to a certain extent, with increases ranging from 7.56–8.54%, 12.05–24.65%, 1.25–4.36% and 3.65–6.68%.

### 3.4. Effects of Attapulgite Application on the Contribution Rate of Organic Carbon, Inorganic Carbon and Total Nitrogen in Aggregates

The contribution rate of attapulgite application to organic carbon, inorganic carbon and total nitrogen to the whole soil in aggregates at each granular level is shown in Table 4. Compared with CK, the application of chemical fertilizer (NPK1) had no significant effect on the contribution rate of organic carbon, inorganic carbon and total nitrogen to whole soil organic carbon, inorganic carbon and total nitrogen in each particles aggregate ($p > 0.05$). The application rate of attapulgite significantly increased the contribution rate of each particles aggregate to whole soil organic carbon, inorganic carbon and total nitrogen, and with the increase in attapulgite application rate, the contribution rate of each particles aggregate gradually increased, reaching a maximum value in NPK1A4 treatment, with an increase range of 0.55–4.46%, 0.20–2.19% and 1.58–6.02%, respectively. The application age of attapulgite has a certain effect on the carbon and nitrogen contribution rate in aeolian sandy soil aggregates.

**Table 4.** Contributions of different particle size aggregates to soil organic carbon, inorganic carbon and total nitrogen.

| Item | | Treatment | Aggregate Size | | | | |
|---|---|---|---|---|---|---|---|
| | | | >2 mm | 2–0.25 mm | 0.25–0.053 mm | <0.053 mm | Sum |
| 2020 (year) | SOC (g kg$^{-1}$) | CK | 6.20 ± 0.37 bc | 6.38 ± 0.40 b | 1.65 ± 0.41 b | 20.08 ± 0.97 b | 34.31 ± 0.54 c |
| | | NPK1 | 5.73 ± 0.26 c | 7.71 ± 0.70 a | 1.19 ± 0.54 c | 21.26 ± 1.88 b | 35.89 ± 0.85 c |
| | | NPK1A1 | 6.55 ± 0.14 b | 7.70 ± 1.02 a | 1.98 ± 0.43 a | 23.46 ± 1.80 a | 39.69 ± 0.84 b |
| | | NPK1A2 | 6.08 ± 0.25 c | 7.10 ± 0.63 a | 2.31 ± 0.24 a | 23.81 ± 0.39 a | 37.04 ± 0.37 b |
| | | NPK1A4 | 6.64 ± 0.29 b | 7.05 ± 0.58 a | 2.13 ± 0.51 a | 21.55 ± 1.46 b | 37.37 ± 0.71 b |
| | | NPK0.8A4 | 7.55 ± 0.11 a | 7.79 ± 0.69 a | 2.20 ± 0.29 a | 24.54 ± 0.72 a | 42.08 ± 0.45 a |
| | SIC (g kg$^{-1}$) | CK | 5.27 ± 0.05 c | 6.87 ± 1.07 c | 3.93 ± 0.33 c | 20.63 ± 2.66 a | 36.7 ± 1.02 c |
| | | NPK1 | 5.06 ± 0.04 c | 7.95 ± 0.49 b | 4.13 ± 0.66 c | 19.23 ± 1.27 a | 36.37 ± 0.62 c |
| | | NPK1A1 | 5.68 ± 0.07 c | 8.91 ± 0.76 a | 5.63 ± 0.23 b | 20.20 ± 0.41 a | 40.42 ± 0.37 b |
| | | NPK1A2 | 5.48 ± 0.18 c | 9.09 ± 0.26 a | 6.58 ± 0.66 a | 21.2 ± 0.39 a | 42.35 ± 0.37 a |
| | | NPK1A4 | 6.52 ± 0.42 b | 9.07 ± 1.58 a | 6.74 ± 0.82 a | 20.59 ± 0.71 a | 42.92 ± 0.88 a |
| | | NPK0.8A4 | 7.11 ± 0.30 a | 9.06 ± 0.49 a | 5.51 ± 1.02 b | 20.83 ± 1.04 a | 42.51 ± 0.71 a |
| | TN (g kg$^{-1}$) | CK | 4.80 ± 0.05 c | 6.79 ± 0.28 b | 2.02 ± 0.28 b | 18.19 ± 2.9 b | 31.8 ± 0.78 d |
| | | NPK1 | 4.61 ± 0.04 c | 7.99 ± 0.46 a | 1.95 ± 0.39 b | 15.75 ± 2.96 b | 30.3 ± 0.96 c |
| | | NPK1A1 | 5.17 ± 0.06 b | 8.17 ± 0.57 a | 3.29 ± 0.27 a | 18.26 ± 1.09 b | 34.89 ± 0.49 c |
| | | NPK1A2 | 4.99 ± 0.17 b | 8.36 ± 0.28 a | 4.08 ± 1.09 a | 24.21 ± 4.64 a | 41.64 ± 1.54 a |
| | | NPK1A4 | 5.93 ± 0.38 a | 8.72 ± 0.39 a | 3.60 ± 0.45 a | 21.11 ± 2.43 a | 39.36 ± 3.65 ab |
| | | NPK0.8A4 | 6.47 ± 0.27 a | 9.01 ± 0.16 a | 2.92 ± 0.92 a | 18.95 ± 2.67 b | 37.35 ± 1.01 b |
| 2021 (year) | SOC (g m$^{-1}$) | CK | 5.58 ± 0.51 c | 7.64 ± 0.42 b | 1.43 ± 0.16 c | 20.21 ± 1.05 b | 34.86 ± 2.14 d |
| | | NPK1 | 4.99 ± 0.32 c | 8.35 ± 0.62 ab | 1.30 ± 0.35 c | 19.04 ± 0.17 b | 33.68 ± 0.36 d |
| | | NPK1A1 | 6.80 ± 0.25 b | 7.58 ± 0.93 ab | 1.85 ± 0.22 b | 22.87 ± 3.05 ab | 39.1 ± 1.13 c |
| | | NPK1A2 | 7.06 ± 0.59 b | 8.21 ± 0.78 ab | 1.83 ± 0.26 a | 25.17 ± 2.35 a | 42.27 ± 0.99 b |
| | | NPK1A4 | 7.03 ± 0.40 b | 8.50 ± 0.85 ab | 2.03 ± 0.22 a | 23.24 ± 0.65 a | 40.80 ± 0.53 c |
| | | NPK0.8A4 | 7.74 ± 0.24 a | 9.73 ± 0.68 a | 2.42 ± 0.39 a | 26.09 ± 1.84 a | 45.98 ± 0.79 a |
| | SIC (g m$^{-1}$) | CK | 5.62 ± 0.64 b | 7.20 ± 1.07 b | 3.76 ± 0.65 c | 25.55 ± 1.94 a | 42.08 ± 1.08 |
| | | NPK1 | 5.25 ± 0.43 b | 7.24 ± 0.94 b | 3.89 ± 0.43 c | 22.88 ± 2.81 a | 39.26 ± 1.15 |
| | | NPK1A1 | 7.04 ± 0.42 a | 7.74 ± 1.61 b | 4.48 ± 0.97 c | 24.39 ± 3.61 a | 43.65 ± 1.65 |
| | | NPK1A2 | 7.38 ± 0.25 a | 7.79 ± 0.51 b | 5.19 ± 0.40 b | 24.20 ± 2.84 a | 44.56 ± 1.01 |
| | | NPK1A4 | 7.10 ± 0.87 a | 7.30 ± 0.72 b | 5.30 ± 0.51 b | 20.67 ± 1.45 b | 40.39 ± 0.88 |
| | | NPK0.8A4 | 7.41 ± 0.21 a | 8.65 ± 0.46 a | 6.35 ± 0.43 a | 20.43 ± 1.50 b | 42.84 ± 0.65 |
| | TN (g m$^{-1}$) | CK | 5.01 ± 0.57 c | 7.00 ± 0.26 b | 2.01 ± 0.39 b | 18.51 ± 3.96 b | 32.53 ± 1.29 |
| | | NPK1 | 4.49 ± 0.46 c | 7.78 ± 0.05 a | 2.33 ± 0.14 b | 18.37 ± 7.20 b | 32.97 ± 1.96 c |
| | | NPK1A1 | 5.94 ± 0.87 c | 6.93 ± 0.61 b | 2.56 ± 0.88 ab | 20.90 ± 1.48 b | 36.33 ± 0.96 b |
| | | NPK1A2 | 6.06 ± 0.97 bc | 8.24 ± 0.20 a | 2.60 ± 0.37 a | 18.86 ± 2.20 b | 35.76 ± 0.93 b |
| | | NPK1A4 | 6.94 ± 0.21 b | 8.87 ± 0.19 a | 2.67 ± 0.60 a | 21.85 ± 5.72 a | 40.33 ± 1.68 a |
| | | NPK0.8A4 | 7.45 ± 0.11 a | 9.56 ± 0.13 a | 3.41 ± 1.35 a | 21.46 ± 5.28 a | 41.88 ± 1.72 a |

Note: SOC: soil organic carbon; SIC: soil inorganic carbon; TN: soil total nitrogen. The value is the mean ± 1 standard deviation of 3 replicate sample values. With different lowercase letters, the difference was statistically significant ($p < 0.05$).

### 3.5. Effects of Changes in Soil Properties on Aggregate Stability and Carbon and Nitrogen Content

Correlation analysis showed that the application of attapulgite soil in 2020 and 2021 showed a significant positive correlation ($p < 0.05$) on the stability of aggregates at various grain grades (>2 mm, 2–0.25 mm, 0.25–0.053 mm and <0.053 mm) in aeolian sandy soil ($p$ 0.05), while EC was negatively correlated with aggregate stability and its organic carbon,

inorganic carbon, total nitrogen content and pH content and overall negative correlation. The effects of attapulgite application on total carbon, organic carbon, inorganic carbon and total nitrogen content in each particles aggregate are shown in Table 5. The application rate of attapulgite soil was positively correlated with the contents of organic carbon, inorganic carbon and total nitrogen in 2–0.25 mm macro-aggregates, 0.25–0.053 mm micro-aggregates and silt + clay particles ($p < 0.05$), indicating that with the increase of the application rate of attapulgite, the contents of organic carbon, inorganic carbon and total nitrogen in 2–0.25 mm macro-aggregates and 0.25–0.053 mm micro-aggregates and silt + clay gradually increased. In contrast, the effect of the high doses of attapulgite soil application on the inorganic carbon and total nitrogen content in the macro-aggregates was not significant.

**Table 5.** Correlation analysis between the amount of attapulgite and various indicators.

| Item | Input | MWD | SOC | SIC | TN | EC |
|------|-------|-----|-----|-----|-----|-----|
| MWD | 0.970 ** | | | | | |
| SOC | 0.678 ** | 0.648 ** | | | | |
| SIC | 0.876 ** | 0.882 ** | 0.652 ** | | | |
| TN | 0.77 ** | 0.21 | −0.06 | 0.27 | | |
| EC | −0.180 | −0.07 | −0.03 | −0.23 | 0.13 | |
| pH | 0.407 | 0.39 | 0.25 | 0.597 ** | 0.21 | −0.20 |

SOC: soil organic carbon, SIC: soil inorganic carbon, TN: total nitrogen, MWD: aggregate stability, EC: electrical conductivity; ** indicates that the correlation is extremely significant ($p < 0.01$).

Through the structural equation model, the influence of attapulgite soil application rate on soil physicochemical properties and soil structure was quantitatively analyzed. Among the many influencing factors, the application rate of attapulgite soil is a key factor in the model, which simulates that after two years of continuous application, the application of attapulgite soil changes the soil physicochemical properties (pH, organic carbon, inorganic carbon, total nitrogen and soil conductivity) and finally improves the stability of soil aggregates (Figure 3). Among them, the path coefficient in the structural equation model can clearly describe the correlation between the factors. The final results showed that the application of attapulgite soil for two consecutive years increased the content of organic carbon, inorganic carbon and clay particles in the whole soil, thereby improving the stability of aggregates.

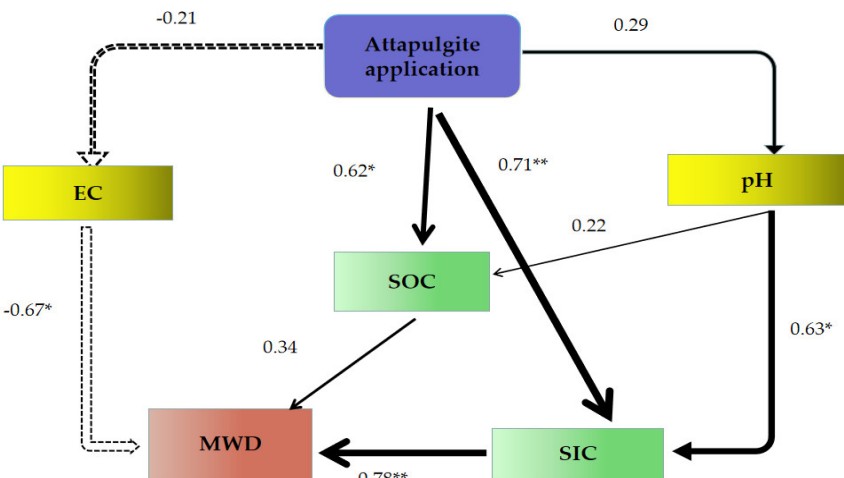

**Figure 3.** Structural equation model explaining the effect of attapulgite application rate on soil physical and chemical properties. Arrows indicate correlation; numbers indicate path coefficients; * indicates that the correlation is significantly different ($p < 0.05$); ** indicates that the correlation is extremely significant ($p < 0.01$). SOC: soil organic carbon, SIC: soil inorganic carbon, EC: Electrical conductivity, MWD: mean weight diameter.

## 4. Discussion

### 4.1. Effects of Attapulgite Application on the Formation and Stability of Sandy Soil Aggregates

The formation process of soil aggregates is influenced by clay minerals, iron–aluminum oxides and organic matter in the soil texture [36]. Rabot et al. [37] showed that soil texture had little effect on soil dry aggregates and hydrostable aggregates in areas with high organic carbon content, while iron oxide played a crucial role in soil hydrostable aggregate formation and stability. Wagner et al. [38] found in the dry region of Australia that agglomerate formation and stability in soils with low sand and organic matter content were mainly affected by clay content. In this study, the application of attapulgite soil increased the content of 0.25–0.053 mm micro-aggregates and the mass fraction of 2–0.25 mm macro-aggregates, which indicated that the application of attapulgite soil was conducive to the formation of micro-aggregates and macro-aggregates and could effectively improve the structure of sandy soil and improve soil quality. This is because attapulgite soil itself has a certain degree of cohesion, which is conducive to aggregating micro-aggregates in sandy soil to form a large aggregate structure. Denef et al. [39] found that in one-to-one clays, the formation of macro-aggregates mainly relies on electrostatic bonding or physical forces, rather than on the action of organic adhesives. Expansive clays such as montmorillonite and other two-to-one clay minerals usually have high cation exchange values and specific surface area, so their own agglomeration is very strong [40]. The microstructure of attapulgite contains many layered structures, and there is a certain gap between these layered structures, forming a large number of micro-pores and micro-pores. These micro-pores and micro-pores make attapulgite have a large specific surface area and rich adsorption sites, which can adsorb and fix various substances [41]. Gao's research results show that the decrease in soil organic matter content leads to the decrease in soil macro-aggregates and the increase in microaggregates, which also indicates that organic carbon is the main substance for the formation of macro-aggregates [42]. Due to the low content of organic matter and clay particles in sandy soil in this experimental field, the high content of soil sand particles and the lack of cementing substances to form macro-aggregates, the application of attapulgite soil increased the content of clay particles in the soil, resulting in the increase in the mass fractions of 0.25–0.053 mm micro-aggregates and 2–0.25 mm macro-aggregates in the soil (Table 2). Suzuki et al. [43] showed that after the application of bentonite on the surface layer of sandy land for 4 years, the content of clay and silt in the surface layer (0–20 cm) of soil increased significantly. On the other hand, attapulgite soil contributes to the improvement of soil water storage capacity, thereby promoting crop growth, and root exudates can promote the formation of macro-aggregates during crop growth [44]. Wu added attapulgite directly to sandy soil; because of its significant gelling characteristics, the dispersed particles can agglomerate to form a larger particle agglomerate, thereby increasing the stability of sandy soil aggregates and improving the water holding capacity of sandy soil [45].

High-valent cations such as calcium, magnesium, ferric and aluminum ions affect the stability of soil aggregates through ionic bond bridges or carbonates, organometallic reactions [46]. Regelink showed that humic acid in bentonite can also promote the formation of large agglomerate structures, especially for moderately weathered soils and some ions in bentonite, such as $Si^{4+}$, $Al^{3+}$, $Ca^{2+}$, $Fe^{3+}$, etc. These ions can promote the formation of aggregates between clay particles and organic particles [47]. After the application of attapulgite to this bentonite, the improvement of the soil environment promotes the growth of various microorganisms, and the secretions of soil microorganisms during the growth process have a cohesive effect on soil particles, especially humic acids and polysaccharides. Daynes et al. [48] found that because microbial cells themselves have a negative charge, they can link soil particles with each other with electrostatic attraction, thereby promoting the formation of macro-aggregates in which fungi can produce hyphae, which can wrap soil particles through mycelium, improve particle viscosity and promote soil aggregate formation. This is consistent with our results—that is, with the increase in attapulgite application, the mass fractions of 2–0.25 mm and 0.25–0.053 mm aggregates in each treated

soil increase with the increase in organic matter content in the whole soil (Figure 3), indicating that the application of attapulgite soil increases soil organic matter content and also promotes the formation of aggregate particles. However, the high application amount of attapulgite had no significant effect on the formation of >2 mm macro-aggregates, which may be that attapulgite clay had limited improvement of organic and inorganic carbon in the whole soil, which in turn limited the formation of macro-aggregates of >2 mm. Zangerlé [49] showed that macro-aggregates were mainly affected by fresh and easily degradable organic matter and temporary organic matter content, and the particle size of aggregates increased with the increase of organic matter content, while the stability of micro-aggregates was mainly affected by refractory organic cemented substances.

*4.2. Effect of Attapulgite Soil Application on Carbon and Nitrogen Content of Sandy Soil Aggregates*

In the absence of exogenous input, soil organic matter formation is a relatively slow process, and the formation mechanism is relatively complex [50]. The main reason for the improvement of soil organic matter by soil amendments without organic matter is the growth of soil microorganisms and plants and promoting the formation of organic matter by changing the physical and chemical properties of soil. In addition to the reasons mentioned above, the organic soil amendments increase soil organic matter content in addition to the other way to directly deliver carbon source substances to the soil to promote the formation of organic matter [51]. Li found that within the first month after adding particulate organic carbon to soil, almost all of the particulate organic carbon in the aggregates was inherently in macro-aggregates, rather than in free-state micro-aggregates or adsorbed in free-state silt + clay particles [52]. Another important factor is the protective effect of silt + clay particles on organic carbon, which can be adsorbed by the clay particles and reduce the mineralization of organic carbon. Cavagnaro et al. [53] showed that soil organic carbon increases with the increase of soil clay content because clay minerals promote the formation of aggregates, which can prevent the decomposition of organic matter through physical protection. The results of this study showed that after the application of high doses (6000 kg·hm$^{-2}$) of attapulgite soil, the contents of organic carbon and total nitrogen in 2–0.25 mm macro-aggregates increased significantly, while there was no significant difference in organic carbon and total nitrogen contents in aggregates in the control and farmland where chemical fertilizers were applied alone. Clay and organic carbon can be combined into organic–inorganic complexes, which will stabilize organic carbon in the soil. At the same time, the maintenance of organic carbon content by clay mainly depends on the type and nature of clay, and mineral clay particles with a crystal structure of two to one can adsorb more organic carbon than mineral clay grains of one to one, because two-to-one clay minerals have stronger adsorption and binder ability to organic matter [54]. In the first year after the application of low amount of attapulgite soil, the change of soil organic carbon was not significant. In the second year of application, the soil organic carbon content increased significantly. After continuous application, macro-aggregates were gradually formed, and the application of attapulgite soil promoted the growth of maize, and the increase in root exudates also contributed to the formation of organic carbon. Chao showed that the leaching loss of nitrogen was reduced, and nitrogen content increased after the application of mineral clay [55]. The results of this study also show that the application of attapulgite increases the organic carbon in the aggregates and the total nitrogen content in the aggregates.

The results of this study showed that the application of high doses of attapulgite significantly increased the content of inorganic carbon in >2 mm macro-aggregates and silt + clay particles, indicating that attapulgite soil not only increased soil organic carbon but also increased the content of inorganic carbon in soil. This may be due to the application of attapulgite soil to increase soil pH, which is conducive to the formation of this biocarbonate in an alkaline soil environment [56]. It can be seen that the application of attapulgite soil induces the production of secondary carbonates, which can reduce the degradation of

organic carbon due to the tight binding of organic matter to soil minerals. The application of attapulgite plays an important role in maintaining the organic carbon content in sandy soil aggregates, which in turn is of great significance for maintaining soil fertility.

The effectiveness of using biochar as a soil amendment may indeed vary depending on specific soil conditions, climate, crop types, and management practices. Soil is a complex ecosystem influenced by multiple factors, so when applying biochar or any soil amendment, the following factors need to be considered: soil type—different soil types have varying textures, structures and chemical properties. The effects of biochar may differ across different soil types, as some soils may exhibit more significant adsorption and aggregation effects with biochar, while others may show weaker effects; climate conditions—climate directly impacts soil moisture, temperature and microbial activity. Under different climate conditions, the performance of biochar in soil improvement may vary; crop type—different crops have distinct soil requirements and adaptability. Some crops may be more sensitive to soil amendments like biochar, while others may show a more subdued response; management practices—soil improvement is a comprehensive process, involving factors such as application rate, frequency, mixing methods, etc. Different management practices can influence the effectiveness of biochar.

Based on the above situation, we recommend the following measures: conduct detailed soil testing and assessment before using biochar to determine the most suitable application methods and rates. At the same time, more methods such as clayey soil blended with calcium lignosulphonate or other materials such as chitosan can be considered to replace attapulgite for improvement [57,58]. Implement sustainable management practices, including appropriate application and regular monitoring, to ensure long-term soil improvement effectiveness and prevent negative environmental impacts. As for resource integration and collaboration, to address cost and supply issues, consider integrating various resources such as government support, farmer cooperatives, research institutions, etc., and seek collaborative partners to jointly advance soil improvement projects. As for water-saving irrigation techniques, when applying biochar, consider adopting water-saving irrigation techniques to optimize water utilization efficiency and reduce water wastage.

## 5. Conclusions

Over the past two consecutive years, the application of high doses (6000 kg·hm$^{-2}$) of bumpy clay (NPK1A4 and NPK0.8A4) has shown significant improvement in sandy soil. The high application of bumpy clay significantly reduced soil bulk density, increased pH value and enhanced the content of physicochemical properties such as total nitrogen, organic carbon and inorganic carbon in the soil. Additionally, the application of high bumpy clay significantly increased the mass fractions of 2–0.25 mm macro-aggregates and 0.25–0.053 mm microaggregates, as well as the content of organic carbon, inorganic carbon and total nitrogen associated with these aggregates, thereby enhancing the stability of soil wind erosion aggregates. On the other hand, the low dose application of bumpy clay (1500–3000 kg·hm$^{-2}$) did not significantly affect the physicochemical properties of the soil. The application of bumpy clay on sandy soil has demonstrated a strong capacity for carbon and nitrogen sequestration, and this effect became more pronounced with time. Consequently, the use of bumpy clay is an effective measure for improving the texture of sandy soil and increasing the carbon and nitrogen sequestration capacity of the soil.

**Author Contributions:** The manuscript was reviewed and approved for publication by all authors. Y.S. conceived and designed the experiments; Z.N. performed the experiments, analyzed the data, drew the figures and wrote the paper; F.A., T.L. and J.L. revised the paper. All authors have read and agreed to the published version of the manuscript.

**Funding:** This research was funded by the National Natural Science Foundation of China (31971730). Qin Chuangyuan Introduced High-level Innovation and Entrepreneurship Talent Project (QCYRCXM-2022-299).

**Institutional Review Board Statement:** Not applicable.

**Informed Consent Statement:** Not applicable.

**Data Availability Statement:** The datasets generated and analyzed during the current study are not publicly available since this experiment was a collaborative effort. The trial data do not belong to me alone but are available from the corresponding author on reasonable request.

**Conflicts of Interest:** The authors declare no conflict of interest.

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
