# Peer review of "Effect of Attapulgite Application on Aggregate Formation and Carbon and Nitrogen Content in Sandy Soil"

_sustainability, doi:10.3390/su151612511_

Round 1

Reviewer 1 Report

The manuscript “Effect of Attapulgite Application on Aggregate Formation and 2 Carbon and Nitrogen Content in Sandy Soil is a very interesting paper which  aims to identify the effects of different application rates of attapulgite on the formation 17 and stability of sandy soil aggregates in northwest China using modern approaches.. The manuscript provides a large amount of data. The manuscript is complete and the different sections of the article are well balanced and adequately supported by the data provided.

The manuscript is clear and answers the objectives. The manuscript is appropriate from the aims and scope of the journal and is potentially publishable in " Sustainability " Journal with some minor changes that authors could add in the manuscript

1.      The introduction section need to improve. Literature from some recent studies published in Tire 1 Journals need to incorporate in the introduction section to attract readers

2.      Hypothesis is missing. Include hypothesis in the introduction section

3.      The quality of Figure 1 is very poor. Map is quite complicated. Redraw simple map using arc GIS with high pixel  to show sampling locations

4.      Method for collection of soil samples should be elaborated in Methodology 

5.      The quality of Figure 2 is poor. Need to improve. Also explain what dose bars in Figure 2 indicates. 

6.      Explain the abbreviations used in Figure 4 in the legend

7.      Discussion should be improved by adding some recent references 

8.      Conclusion need to improve. Must be based on your own findings.

9.      The whole article need to be checked for typos and grammar error

10.  All reference should be according to journal guidelines

1.     

Author Response

Thank you very much for your great efforts for dealing with our manuscript titled 'Effect of Attapulgite Application on Aggregate Formation and Carbon and Nitrogen Content in Sandy Soil', and we greatly appreciate the reviewers and editors for their constructive comments and suggestions. These comments and suggestions are very valuable and helpful for improving the quality of our paper. We have considered all the comments and suggestions carefully and made revisions of each point. These revisions are highlighted in red in our revised manuscript. Our responses to the comments and suggestions as well as detailed changes in the context are summarized as follows.

Reviewer 1:

Point 1: The introduction section need to improve. Literature from some recent studies published in Tire 1 Journals need to incorporate in the introduction section to attract readers.

Response 1:In the introduction, we have added research progress in recent years and references:"It was found that the biological crust succession had significant influence on the stability of surface soil aggregates during the ecological project of "returning farmland to forest" on the Loess Plateau[18]. Wei's results show that the increase of soil aggregates during vegetation restoration in karst areas is significantly correlated with the increase of nutrient content[19].” (line77-83).

Point 2: Hypothesis is missing. Include hypothesis in the introduction section.

Response 2: We've added hypotheses to the introduction: “Our hypothesis is that improving sandy soil by applying attapulgite clay can effectively improve soil texture and structure, thereby increasing soil carbon and nitrogen fixation capacity.” (line 120-123).

Point 3: The quality of Figure 1 is very poor. Map is quite complicated. Redraw simple map using arc GIS with high pixel  to show sampling locations.

Response 3: We have re-produced the general map of the study area and improved the clarity of the picture.(line 143)

Point 4: Method for collection of soil samples should be elaborated in Methodology.

Response 4: Our methods of soil sampling are described in detail: “For each field, the plum blossom method was used for sampling. First, the soil surface was cleaned, and then the undisturbed soil was removed with a small shovel. During the sampling process, the undisturbed soil should avoid extrusion deformation, and then the soil clumps at five sample sites were mixed.” (line 190-193).

Point 5: The quality of Figure 2 is poor. Need to improve. Also explain what dose bars in Figure 2 indicates.

Response 5: We reworked the images, improved the clarity of the images, and added annotations to the images.(line 314-322).

Point 6: Explain the abbreviations used in Figure 4 in the legend.

Response 6: We have added explanations to the images.(line372-373).

Point 7: Discussion should be improved by adding some recent references.

Response 7: We have updated part of the literature and added corresponding explanations. “We have updated part of the literature and added corresponding explanations” (line 404-411). “The effectiveness of using biochar as a soil amendment may indeed vary depending on specific soil conditions, climate, crop types, and management practices. Soil is a complex ecosystem influenced by multiple factors, so when applying biochar or any soil amendment, the following factors need to be considered:Soil Type: Different soil types have varying textures, structures, and chemical properties. The effects of biochar may differ across different soil types, as some soils may exhibit more significant adsorption and aggregation effects with biochar, while others may show weaker effects.Climate Conditions: Climate directly impacts soil moisture, temperature, and microbial activity. Under different climate conditions, the performance of biochar in soil improvement may vary. Crop Type: Different crops have distinct soil requirements and adaptability. Some crops may be more sensitive to soil amendments like biochar, while others may show a more subdued response. Management Practices: Soil improvement is a comprehensive process, involving factors such as application rate, frequency, mixing methods, etc. Different management practices can influence the effectiveness of biochar.

Based on the above situation, we recommend the following measures: conduct detailed soil testing and assessment before using biochar to determine the most suitable application methods and rates. At the same time, more methods such as clayey soil blended with calcium lignosulphonate or other materials such as chitosan can be considered to replace attapulgite for improvement[57,58]. Implement sustainable management practices, including appropriate application and regular monitoring, to ensure long-term soil improvement effectiveness and prevent negative environmental impacts. Resource Integration and Collaboration: to address cost and supply issues, consider integrating various resources such as government support, farmer cooperatives, research institutions, etc., and seek collaborative partners to jointly advance soil improvement projects. Water-Saving Irrigation Techniques: When applying biochar, consider adopting water-saving irrigation techniques to optimize water utilization efficiency and reduce water wastage.” (line 499-525)

Point 8: Conclusion need to improve. Must be based on your own findings.

回应8:我们重新总结了结论:“连续两年,高剂量(6000 kg·hm-2)凹凸不平的粘土(NPK1A4和NPK0.8A4)在砂质土壤中的应用显示出显着改善。高施用凹凸不平的黏土显著降低了土壤容重,提高了pH值,提高了土壤中全氮、有机碳、无机碳等理化性质的含量。此外,施用高凹凸不平的黏土显著提高了2—0.25 mm大团聚体和0.25-0.053 mm微团聚体的质量分数,以及这些团聚体伴生的有机碳、无机碳和全氮含量,从而提高了土壤风蚀团聚体的稳定性。另一方面,低剂量施用凹凸不平的黏土(1500-3000 kg·hm-2)对土体理化性质影响不显著。在沙质土壤上施用凹凸不平的粘土显示出很强的碳氮固存能力,并且这种效果随着时间的推移变得更加明显。因此,使用凹凸不平的粘土是改善砂质土壤质地、提高土壤碳氮固存能力的有效措施。(第 527-541 行)。

9整篇文章需要检查是否有错别字和语法错误。

答复9:对语法问题进行了检查和修改。

要点10所有参考文献应符合期刊指南

答复10:审查和修改了参考文献问题。

我们非常感谢您和审稿人对我们论文的意见和建议。

期待收到您的来信。

谢谢你,最诚挚的问候。

你的真诚,

牛子茹

通讯作者:

姓名:苏永忠

你的真诚,

牛子茹

通讯作者:

姓名:苏永忠

Reviewer 2 Report

The manuscript deals with applying Attapulgite (clay mineral that has been used in various agricultural applications, including soil amendment) on the aggregate formation and Carbon and Nitrogen contents enhancement in sandy soil. The work has practical relevance and is suitable for publication in the sustainability journal as the scope falls within the journal. However, there are severe limitations noted in the study and the authors are required to address the following queries while revising their manuscript:

1.       What is the rationale behind choosing sandy soil in the present study? Is the study region specific? Or the findings of the study will be applicable to other soil types which have finer fractions? The authors must address this issue very carefully relying on the existing literature while revising the manuscript.

2.       While Attapulgite can offer benefits in certain contexts, there are several potential shortcomings when applying Attapulgite on aggregate formation and carbon and nitrogen content in sandy soil. How does addition of Attapulgite compare to existing strategies of enhancing the organic uptake capacity of sandy soils? Provide substantial references for the claims being made to support the statement.

3.       Attapulgite itself does not contain significant amounts of essential plant nutrients such as carbon and nitrogen. While it can help retain moisture and improve soil structure, it does not contribute to the nutrient content of the soil. Therefore, relying solely on Attapulgite may not address the nutrient deficiencies in sandy soils.

4.       Organic matter is crucial for improving soil fertility and enhancing carbon and nitrogen content. Attapulgite, being a mineral material, does not provide organic matter to the soil. Sandy soils typically lack organic matter, and simply applying Attapulgite may not sufficiently increase the carbon and nitrogen content necessary for healthy plant growth. The authors must carefully address this issue while revising the manuscript by providing valid justifications from the existing literature.

5.       Attapulgite can have an impact on soil pH depending on its specific properties and the initial pH of the soil. Sandy soils are often characterized by being acidic, and adding Attapulgite may further decrease the pH. Ironically in the present study, following 2 years of application, the pH of the sandy soil increased (towards alkaline) which is contradictory to theoretical aspects. The authors must explain this anomaly. Reduced pH can negatively affect nutrient availability and overall soil health if not properly managed. How do the authors address this issue? Provide valid critique relying on existing literature and support your research findings reported in the study.

6.       Sandy soils have a high drainage capacity due to their coarse texture. Attapulgite, being a clay mineral, has the potential to increase water retention in sandy soils. While this can be advantageous in some cases, excessive water retention can lead to poor drainage and increased risks of waterlogging, which can adversely affect plant growth.

7.       Attapulgite may not be readily available or affordable in certain regions. The cost-effectiveness of using Attapulgite as a soil amendment needs to be considered, especially when alternative organic amendments or fertilizers may be more suitable and cost-efficient for improving soil fertility and carbon and nitrogen content. Suppose if the area of 200 acres has to be improved by the addition of Attapulgite, what are the associated carbon emissions and how do they compare with existing techniques to enrich the nutrient uptake capacity of the soil in similar lines? The authors are strongly encouraged to refer to the following articles

(https://doi.org/10.3389/fenvs.2023.1214988; https://doi.org/10.3390/su15076117) and provide meaningful inferences while revising the manuscript.

8.       It's important to note that the effectiveness of Attapulgite as a soil amendment can vary depending on specific soil conditions, climate, crop type, and management practices. Before applying any soil amendment, including Attapulgite, it is advisable to carry out some elementary tests. Provide these aspects while revising the manuscript.

9.       What are the practical difficulties that may arise while applying Attapulgite in real field conditions? What suggestions the authors provide to overcome this issue.

10.   The entire conclusions section has to be redrafted focusing on the qualitative and quantitative aspects of the study.

The authors must provide a strong response sheet for each of the queries raised above. Since the experiments were conducted for 2 years, the reviewer sees some merit in the manuscript and hence urges the authors to take the suggestions seriously and address them. 

Author Response

Thank you very much for your great efforts for dealing with our manuscript titled 'Effect of Attapulgite Application on Aggregate Formation and Carbon and Nitrogen Content in Sandy Soil', and we greatly appreciate the reviewers and editors for their constructive comments and suggestions. These comments and suggestions are very valuable and helpful for improving the quality of our paper. We have considered all the comments and suggestions carefully and made revisions of each point. These revisions are highlighted in red in our revised manuscript. Our responses to the comments and suggestions as well as detailed changes in the context are summarized as follows.

Reviewer 2:

Point 1: What is the rationale behind choosing sandy soil in the present study? Is the study region specific? Or the findings of the study will be applicable to other soil types which have finer fractions? The authors must address this issue very carefully relying on the existing literature while revising the manuscript.

Response 1:We have re-added the relevant content in the introduction: “Aeolian sand is widely distributed in northwest China, with low organic matter content and difficulty to accumulate, high sand content, poor soil structure, and general lack of nutrients. The improvement of soil fertility and carbon fixation capacity has become the top priority of local agricultural development[28].”(line 105-108).

Point 2: While Attapulgite can offer benefits in certain contexts, there are several potential shortcomings when applying Attapulgite on aggregate formation and carbon and nitrogen content in sandy soil. How does addition of Attapulgite compare to existing strategies of enhancing the organic uptake capacity of sandy soils? Provide substantial references for the claims being made to support the statement.

Response 2: We have re-added the relevant content in the introduction: “In recent decades, due to the limited application amount of organic fertilizer and the reduction of surface water irrigation area, the input of soil fine matter is blocked. Under this background, the newly reclaimed sandy soil structure is retarded, and the soil texture becomes more loose, which leads to the decline of its water and fertilizer retention ability, affecting the normal growth and yield of crops. In this context, effective soil improvement measures, such as the rational use of soil amendments such as attapulgite, can help to promote the formation of soil structure, improve soil fertility and water retention, thereby increasing crop yield and achieving sustainable agricultural development[30]” (line 110-118)

Point 3: Attapulgite itself does not contain significant amounts of essential plant nutrients such as carbon and nitrogen. While it can help retain moisture and improve soil structure, it does not contribute to the nutrient content of the soil. Therefore, relying solely on Attapulgite may not address the nutrient deficiencies in sandy soils.

Response 3: In the introduction, the characteristics of attapulgite are introduced in detail: “Attapulgite clay minerals can play a good role as nutrient carriers in soil. This is mainly due to the adsorbability and ion exchange capacity of clay minerals, allowing them to effectively adsorb and retain nutrients in the soil, including cations and anions. This adsorption and ion exchange ability enables clay minerals to act as a stable reservoir of nutrients in the soil, reducing nutrient loss and leaching, which is conducive to the absorption and utilization of nutrients by plants[12].”( line 61-67).

Point 4: Organic matter is crucial for improving soil fertility and enhancing carbon and nitrogen content. Attapulgite, being a mineral material, does not provide organic matter to the soil. Sandy soils typically lack organic matter, and simply applying Attapulgite may not sufficiently increase the carbon and nitrogen content necessary for healthy plant growth. The authors must carefully address this issue while revising the manuscript by providing valid justifications from the existing literature.

Response 4: We noted this in our materials and methods: “The conventional fertilization level is N-P2O5-K2O, 320-240-200 kg·hm-2; The 20 % reduction of fertilizer application level is N-P2O5-K2O, 256-192-160 kg·hm-2. The fertilizer applied is attapulgite based compound fertilizer, which is granulated with attapulgite clay mixed with urea, diammonium phosphate and potassium sulfate in a certain proportion. The nutrient content is N-P2O5-K2O 17.8%-16.1%-10%, containing 20 % attapulgite clay, and the total nutrient content is 43.9 %. The organic carbon content of attapulgite clay minerals is 5.43g·kg-1, the inorganic carbon content is 6.71g·kg-1, the total nitrogen content is 0.67g·kg-1, and the pH is 7.9. One-time light and simplified fertilization method is adopted, all of which are used for basic application. Attapulgite based compound fertilizer and urea are combined to provide the required N, P and K total nutrients.”( line 168-178).

Point 5: Attapulgite can have an impact on soil pH depending on its specific properties and the initial pH of the soil. Sandy soils are often characterized by being acidic, and adding Attapulgite may further decrease the pH. Ironically in the present study, following 2 years of application, the pH of the sandy soil increased (towards alkaline) which is contradictory to theoretical aspects. The authors must explain this anomaly. Reduced pH can negatively affect nutrient availability and overall soil health if not properly managed. How do the authors address this issue? Provide valid critique relying on existing literature and support your research findings reported in the study.

Response 5: In the arid area of northwest China, the climate is mainly arid and semi-arid, with little precipitation and strong evaporation. Under drought conditions, the salt in the soil is relatively easy to accumulate. As the water evaporates, the concentration of salt in the soil gradually increases, causing the soil to alkalize. We have detailed the basic physical and chemical properties of the original soil in the materials and methods, and the original pH content of the soil is 8.9-9.15. Generally speaking, soil pH content has a strong correlation with the inorganic carbon content in soil. In alkaline environment, carbonate is easy to form, and attapulgite clay minerals are easy to form inorganic carbon, thereby increasing the total carbon content of soil. (Rong Y, Su Y, Tao W, et al. Effect of chemical and organic fertilization on soil carbon and nitrogen accumulation in a newly cultivated farmland. Journal of integrative agriculture, 15(3), 658-666(2016). )

Point 6: Sandy soils have a high drainage capacity due to their coarse texture. Attapulgite, being a clay mineral, has the potential to increase water retention in sandy soils. While this can be advantageous in some cases, excessive water retention can lead to poor drainage and increased risks of waterlogging, which can adversely affect plant growth.

Response 6: Thank you very much for your suggestion. The main type of soil applied in our paper is aeolian sand soil, and one of the disadvantages of aeolian sand soil is that it has too strong drainage ability and poor water and fertilizer retention ability. In arid areas, due to the lack of water resources, the aeolian sand is not conducive to the sustainable development of local agriculture. So we propose to use attapulgite to improve the water and fertilizer retention ability of aeolian sand soil and improve the water and fertilizer utilization efficiency. (Su YZ, Yang R, Liu WJ. Effects of Agricultural Management Practices on Soil Organic Carbon and Its Fractions in Newly Cultivated Sandy Soil in Northwest China. Scientia Agricultura Sinica , 45(14), 2867-2876(2012).)

Point 7: Attapulgite may not be readily available or affordable in certain regions. The cost-effectiveness of using Attapulgite as a soil amendment needs to be considered, especially when alternative organic amendments or fertilizers may be more suitable and cost-efficient for improving soil fertility and carbon and nitrogen content. Suppose if the area of 200 acres has to be improved by the addition of Attapulgite, what are the associated carbon emissions and how do they compare with existing techniques to enrich the nutrient uptake capacity of the soil in similar lines? The authors are strongly encouraged to refer to the following articles(https://doi.org/10.3389/fenvs.2023.1214988;https://doi.org/10.3390/su15076117) and provide meaningful inferences while revising the manuscript.

Response 7: We added relevant content to the discussion and cited relevant literature.(line513-525).

Point 8: It's important to note that the effectiveness of Attapulgite as a soil amendment can vary depending on specific soil conditions, climate, crop type, and management practices. Before applying any soil amendment, including Attapulgite, it is advisable to carry out some elementary tests. Provide these aspects while revising the manuscript.

Response 8: We added data to the methodology (line 168-178) and added relevant content to the discussion: “The effectiveness of using biochar as a soil amendment may indeed vary depending on specific soil conditions, climate, crop types, and management practices. Soil is a complex ecosystem influenced by multiple factors, so when applying biochar or any soil amendment, the following factors need to be considered:Soil Type: Different soil types have varying textures, structures, and chemical properties. The effects of biochar may differ across different soil types, as some soils may exhibit more significant adsorption and aggregation effects with biochar, while others may show weaker effects.Climate Conditions: Climate directly impacts soil moisture, temperature, and microbial activity. Under different climate conditions, the performance of biochar in soil improvement may vary. Crop Type: Different crops have distinct soil requirements and adaptability. Some crops may be more sensitive to soil amendments like biochar, while others may show a more subdued response. Management Practices: Soil improvement is a comprehensive process, involving factors such as application rate, frequency, mixing methods, etc. Different management practices can influence the effectiveness of biochar.” (line499-512).

Point 9: What are the practical difficulties that may arise while applying Attapulgite in real field conditions? What suggestions the authors provide to overcome this issue.

Response 9: We added relevant content to the discussion: “Based on the above situation, we recommend the following measures: conduct detailed soil testing and assessment before using biochar to determine the most suitable application methods and rates. At the same time, more methods such as clayey soil blended with calcium lignosulphonate or other materials such as chitosan can be considered to replace attapulgite for improvement[57,58]. Implement sustainable management practices, including appropriate application and regular monitoring, to ensure long-term soil improvement effectiveness and prevent negative environmental impacts. Resource Integration and Collaboration: to address cost and supply issues, consider integrating various resources such as government support, farmer cooperatives, research institutions, etc., and seek collaborative partners to jointly advance soil improvement projects. Water-Saving Irrigation Techniques: When applying biochar, consider adopting water-saving irrigation techniques to optimize water utilization efficiency and reduce water wastage.” (line513-525).

Point 10: The entire conclusions section has to be redrafted focusing on the qualitative and quantitative aspects of the study.

Response 10: We re-summarize the conclusion: “Over the past two consecutive years, the application of high doses (6000 kg·hm-2) of bumpy clay (NPK1A4 and NPK0.8A4) has shown significant improvement in sandy soil. The high application of bumpy clay significantly reduced soil bulk density, increased pH value, and enhanced the content of physicochemical properties such as total nitrogen, organic carbon, and inorganic carbon in the soil. Additionally, the application of high bumpy clay significantly increased the mass fractions of 2-0.25 mm macro-aggregates and 0.25-0.053 mm microaggregates, as well as the content of organic carbon, inorganic carbon, and total nitrogen associated with these aggregates, thereby enhancing the stability of soil wind erosion aggregates. On the other hand, the low dose application of bumpy clay (1500-3000 kg·hm-2) did not significantly affect the physicochemical properties of the soil. The application of bumpy clay on sandy soil has demonstrated a strong capacity for carbon and nitrogen sequestration, and this effect becomes more pronounced with time. Consequently, the use of bumpy clay is an effective measure for improving the texture of sandy soil and increasing the carbon and nitrogen sequestration capacity of the soil.” (line 527-541).

We would like to express our great thanks to you and the reviewers for the comments and suggestions on our paper.

Looking forward to hearing from you.

Thank you and best regards.

Yours sincerely,

Ziru Niu

Name: Yongzhong Su

Round 2

Reviewer 2 Report

The authors have addressed my comments raised during the first review. The reviewer is satisfied with their response and recommends acceptance of the revised version of manuscript.